# Age-Friendly Ecosystems: Expert Voices from the Field

**DOI:** 10.3390/geriatrics8040068

**Published:** 2023-06-21

**Authors:** Terry Fulmer, Kim Dash, Jody Shue, JiHo Chang, Jessica Huang, Abby Maglich

**Affiliations:** 1The John A. Hartford Foundation, New York, NY 10022, USA; jiho.chang@johnahartford.org (J.C.); jessica.huang@johnahartford.org (J.H.); abby.maglich@johnahartford.org (A.M.); 2Education Development Center, Waltham, MA 02451, USA; kdash@edc.org; 3School of Public Health, Boston University, Boston, MA 02118, USA; jshue@agefriendly.org; 4School of Public Health, Columbia University Mailman, New York, NY 10032, USA

**Keywords:** age-friendly, ecosystems, aging, measurement

## Abstract

(1) Background: With the growth of the age-friendly movement, age-friendly ecosystems (AFE) garnered more attention. The successful development of an AFE is contingent on unified efforts across different stakeholders; however, limited efforts were made to help create a common understanding of the necessary components of an AFE. (2) Methodology: In response, The John A. Hartford Foundation and The Age-Friendly Institute hosted a series of convenings of international experts to identify a working definition of the characteristics composing an AFE. The goal of these convenings was to provide a foundation on which to unite cross-sector age-friendly work. (3) Results: This paper discussed the findings of the convenings and provided a framework from which future age-friendly work must draw upon. (4) Conclusions: This paper presented a necessary change in how we conceive AFEs.

## 1. Defining an Age-Friendly Ecosystem

The growth in scholarship, civic planning, and programming in age-friendly initiatives led to progress across various sectors, but currently, the efforts were mainly conducted independently of each other. A coordinated effort across age-friendly sectors (health systems, cities/communities, universities, public health, and employers) to operationalize an age-friendly ecosystem (AFE) began to advance the existing work and to work across the various spheres of the AFE. This effort aims to address the comprehensive needs of all older adults and achieve a broader, collective, and long-lasting impact. An age-friendly ecosystem is defined as a comprehensive, collectively built, ever-expanding platform whose goal is to improve quality of life for older adults around the world through enhanced, collaborative impact. The age-friendly ecosystem does not propose a hierarchy of any particular approach to improving quality of life for older adults. Rather, this platform acknowledges the unique strengths and contributions of existing approaches and promotes enhanced continuity and collective impact across settings [1]. The purpose of this paper is to document the voices of 57 international experts who came together to achieve agreement on a series of six characteristics that define an AFE [2]. We previously published a technical report that described the six characteristics. In this paper, we elucidate the voices of the experts in order to characterize the features of importance to the varied stakeholders [1]. All the invited experts agreed that it is valuable to build on the existing work started in multiple sectors and countries worldwide, based on the premise that an AFE is required to achieve well-being for older adults globally (expert participants are listed in Appendix A). By communicating across sites where older adults live, work, and receive health care and caregiving, all of us are strengthened. The collaboration aims to design and implement a shared language and metrics that can help ensure inclusivity and well-being as determined by individual biology, personal choices, relationships with others, home settings, neighborhoods, healthcare settings, and workplaces [3].

## 2. Understanding of Age-Friendly Ecosystems

The age-friendly ecosystem comprises public health systems, cities and communities, health systems, universities, and employers [1]. Each of these sectors currently follows a guiding “age-friendly” framework. Understanding shared characteristics across these frameworks will help unite work and move towards an AFE. The COVID-19 pandemic spotlighted the pitfalls in our current approach toward older adult well-being and provided an environment to encourage more stakeholders to incorporate age-friendly approaches [4]. The overarching goal of an age-friendly ecosystem is to ensure a collective effort across the various sectors (Figure 1) and engage stakeholders who can shape the ultimate goal of improving the quality of life for older adults. In creating the AFE, a common language, shared metrics can help guide collaboration across sectors [2]. The AFE is crucial because it enhances individual efforts and creates a greater collective impact. Now more than ever, it is clear that the impact of the environment on older adults, specifically on their health and well-being, must be considered. Developing a shared definition and working to create an AFE can ameliorate our current state of discoordination and improve the lives of older adults, their families, and their caretakers.

## 3. Defining the Age-Friendly Ecosystem

The importance of AFEs is becoming more widely accepted, but there is still great potential for collaboration across sectors. Most age-friendly work was conducted on an organizational level, with individual hospitals, health systems, and communities incorporating age-friendly approaches [4]. Synergies across different institutions are required to continue building momentum; however, this can only be effectively undertaken by first embracing a shared language and understanding the characteristics that compose an AFE [2,6].

## 4. The Approach

The need described above led to a series of convenings involving 57 international experts to determine a common understanding of the definition and the shared characteristics of an AFE which were described previously [1]. Creating that shared language facilitates partnerships across sectors to bolster the efficacy of different programs, and identifies standardized measures to assess community needs, efficacy, and outcomes [7]. These measures help ensure that programs are evidence-based and linked to positive outcomes and meaningful change. This is a crucial next step as it will help scale up ongoing work. To begin the collaboration, 44 international experts joined together to use an expert panel approach to address the aim.

## 5. Methodology

The John A. Hartford Foundation and The Age-Friendly Institute hosted a series of three conventions of national and international experts to discuss AFE definition and characteristics. Experts convened virtually to review characteristics identified via a stakeholder survey, literature review, synthesis, and thematic analysis. Fifty-seven leaders of organizations representing educational, employment, healthcare, and urban and regional planning sectors agreed to participate, including the heads of influential private and corporate foundations, international and national non-governmental organizations, government agencies, academic institutions, and healthcare organizations. A Delphi technique, also known as estimate–talk–estimate approach was used to reach saturation [1].

In anticipation of the first session, participating experts reviewed the characteristics and supporting practices gleaned from the literature that comprise AFEs prior to the convening (please see Appendix B).

Session 1, held in December 2020, was opened by asking participants to share words they most associate with the term “age-friendly” (Figure 2).

Experts were then divided into four groups with representatives of different sectors included in each of the four groups. All breakout sessions were audio-recorded and note takers documented major themes that group facilitators presented back to the larger group for discussion and comment. Summary reports from the session and more detailed information about the characteristics and associated measures can be found on the John A. Hartford Foundation website (https://www.johnahartford.org/grants-strategy/current-strategies/age-friendly/age-friendly-ecosystem, accessed on 24 February 2023). During the first small group breakout, trained facilitators asked group members to address three main questions after having reviewed methods used to derive initial characteristics or ways of describing an AFE: Are these the best characteristics to describe an AFE? Why or why not? Tell us how you think the characteristics work across initiatives (your own work and that of others). Are there characteristics that we are missing? During the second small group breakout, participants indicated which of the following actions would have the greatest impact on development of a working AFE: identifying where we have the most in common to overcome separated approaches to our work; overcoming fears that an AFE will add an additional layer to our work; identifying foundation and government support to address payment barriers to achieving the work; encouraging additional major leadership from groups such as WHO, AARP, and others to tackle policy barriers; and demonstrating value, cost savings, and efficiency to overcome inertia.

Following the first session, a synthesis of learnings was conducted which surfaced updates to the first draft of shared characteristics based on collective thinking across fields. The updated characteristics of an AFE are outlined in Table 1 below. The updated shared characteristics of an AFE were shared with expert participants in advance of the second session.

The goal of the second session, held in March 2021, was to build upon our work in December by exploring areas for collaboration across sectors, beginning to identify measures that can be aligned across age friendly settings, and developing shared understandings of an age-friendly ecosystem in order that it can become an actionable roadmap for practitioners. The second session began with participants responding to the question “What will be the number one benefit that will be achieved by organizations by becoming part of an age-friendly ecosystem?”. Responses from attendees can be found in Table 2 below.

Following this, participants were divided into virtual breakout rooms and worked together to identify the most impactful actions for building an AFE. They were asked to reflect on the following questions: Which goal (in each of the 6 characteristics) do you think is the top priority? Do you agree with the survey results? How can we find the best opportunities to collaborate based upon your priorities? If you had to choose one of the six characteristics of an AFE to explore more deeply through a discussion of goals and measures of impact, which would you choose?

## 6. Expert Voices Defining an Age-Friendly Ecosystem

The purpose of this endeavor was to seed collective action in the AFE and ask leaders and adherents of each framework to consider a collective approach to improving older adult well-being. Experts conceded that a first, critical step in the process was to identify commonalities across sectors of work, noting that once they identified commonalities, they would be able to begin to overcome inertia, policy, and payment barriers. However, despite efforts in different sectors and various social-ecological levels, there is a need to expand the analysis across the multiple age-friendly frameworks to determine what they have in common. This group did not try to create a new age-friendly framework. Instead, through a carefully structured process, those representing all age-friendly sectors focused on establishing a set of characteristics that defines the AFE. This was considered a first step in working toward collective impact, followed by goal setting, measurement, and action planning.

Drawing from the social-ecological model [8], experts described how age-friendly characteristics apply to their work and across age-friendly frameworks. This model acknowledged the interplay between older adults’ biology and behaviors, their social ties, and their environments. Social-ecological models recognize that individuals are influenced by the people and environments around them, such as social norms, and the environment can be used to target health behaviors (Figure 3) [8,9]. Moreover, social-ecological models also tend to be developmental, asserting that relationships and contexts affect how individuals age.

## 7. Expert Viewpoints

The sessions resulted in a rich discussion regarding both the value and the challenges of achieving the aim of a shared language and framework. One participant described the work of her institution’s integrated community and mobile health service team. Part of their function was to monitor health indicators and social determinants likely to affect health, such as environmental factors that contribute to fall risk and food kept in home pantries. She explained, “The feedback that we are gathering from these visits is [that] there’s a lot of support services that [older adults] need.” Another participant, based at a large urban teaching hospital, described how she worked at the community and individual levels to promote change. Her hospital was an age-friendly health system that engaged communities and community-based organizations through formal discharge planning. Staff worked with patients to design discharge plans so that when patients left the hospital, they had a written plan that can be implemented or supported by some of the hospitals’ affiliated community-based organizations. Two graphics (Figure 4 and Figure 5) presented at the March 2021 convention provide a visual example of the stark difference in older adults’ experiences within a non-age-friendly ecosystem and within an age-friendly ecosystem. The graphics used the example of vaccinations to exemplify how AFEs in action can substantially improve older adults’ experience when seeking care by making it more seamless, caring, and understanding.

Experts indicated that it is essential that sectors not only respond to what older adults want and need but also have them plan and develop the services and products they use. One meeting participant described how co-designing programming is responsive to older adults’ needs by engaging them in designing materials that promote shared decision-making with their healthcare providers. This participant indicated that her team co-designed educational materials with focus groups of older adults who provided ample feedback around language and formatting. Her team was discovering that upfront engagement and education were more effective in promoting behavior change among older adults than trying to influence providers to initiate and encourage older adult behavior change. She notes, “If we’re engaging older adults [through information sharing and education] … we’ve got an older adult coming in and saying: Hey, doc, I want you to listen to what matters to me, [then] that conversation is much more likely to happen than if we try to tell the docs: Hey, ask your patients what matters to them”.

Others emphasize the importance of reaching people at different points in their lives, promoting intergenerational arrangements, and directly addressing ageism. One participant noted that the various age-friendly frameworks inform and enhance each other. She explained, “So what’s happening on college campuses is [that we are] teaching college kids how to think and act and basically to be age-less in their mind. Similarly, health care is coming at it from a different direction and employment [from another]. [We’re] reaching people at different points in their lives where [we are] calling attention to a bias that shouldn’t exist or should be better understood.” She went on to say, “In the university sector, one of the reasons [her] university developed this [age-friendly programming] was because we knew we had a perfect opportunity to talk to students about how they are going to age and involve them in aging [research]”.

So that age-friendly systems can be better realized, training in essential competencies was also viewed as critical. “If we don’t have adequate and accurate knowledge about how best to interact with older patients, older clients, older residents, we can do a lot of damage.” Another noted, “we’re talking about lifelong learning, helping people to age, but also to have people that will assist them in the health sector.” One expert representing an Age-Friendly University (AFU) indicated that training is interdisciplinary and well-integrated into the academic experience with opportunities for real-world practice: “as an academic and a practitioner, we’ve been aligning our age-friendly university, which is an inter-professional campus, and realizing that we need to have sites for our workforce to develop.” At this AFU, gerontology principles are integrated into multiple undergraduate programs and curricula, interdisciplinary graduate-level coursework, and community-based practicums.

## 8. Summary

Through these conventions, age-friendly leaders were reminded of the importance of a collective impact. According to Kania and Kramer, a collective impact can be described as the commitment of practitioners and stakeholders from different sectors to a common agenda for solving a complex social challenge [10]. As we continue to make progress with our roadmap to an AFE, it is important to maintain the underlying values identified through these discussions. These include bringing in older adults when creating initiatives, empowering them as local champions, considering the diverse needs across groups of older adults, and, most importantly, maintaining respect for all individuals.

In this paper, we provided a narrative from the experts as they considered the opportunities, values, and challenges of implementing a structure that can lead to policies and practice change in how we think about age-friendly ecosystems globally. With the aging of the population and the improved health and well-being for many as they reach their older age, everything we can do to improve our clarity of purpose and continuity of language across the multiple sectors that impact the well-being of older adults will do much to enhance the experience of older age. In the future, as teams collaborate, we will have the experience and data to guide us further into the ecosystem that will inform policy changes and accelerate the coordination that can best serve society. In order to create lasting solutions at scale, they argue, practitioners and stakeholders of all types need to coordinate their efforts and work together around clearly defined goals and a shared vocabulary to describe what it means to be age-friendly, regardless of setting. The involvement of expert voices in the development of shared language to describe the AFE meaningfully addressed this challenge.

To continue to move this work forward, a coordinating backbone agency with a national footprint would be in a position to foster and manage cross-sector initiatives with the skills and resources to convene, build trust, provide tools, and shepherd meaningful solutions.

## Figures and Tables

**Figure 1 geriatrics-08-00068-f001:**
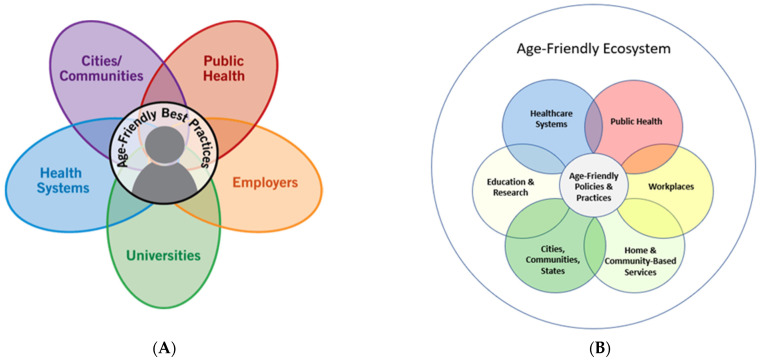
(**A**) Evolving Age-Friendly Ecosystem Sectors [5] IHI 2023. (**B**) Evolving Age-Friendly Ecosystem Sectors.

**Figure 2 geriatrics-08-00068-f002:**
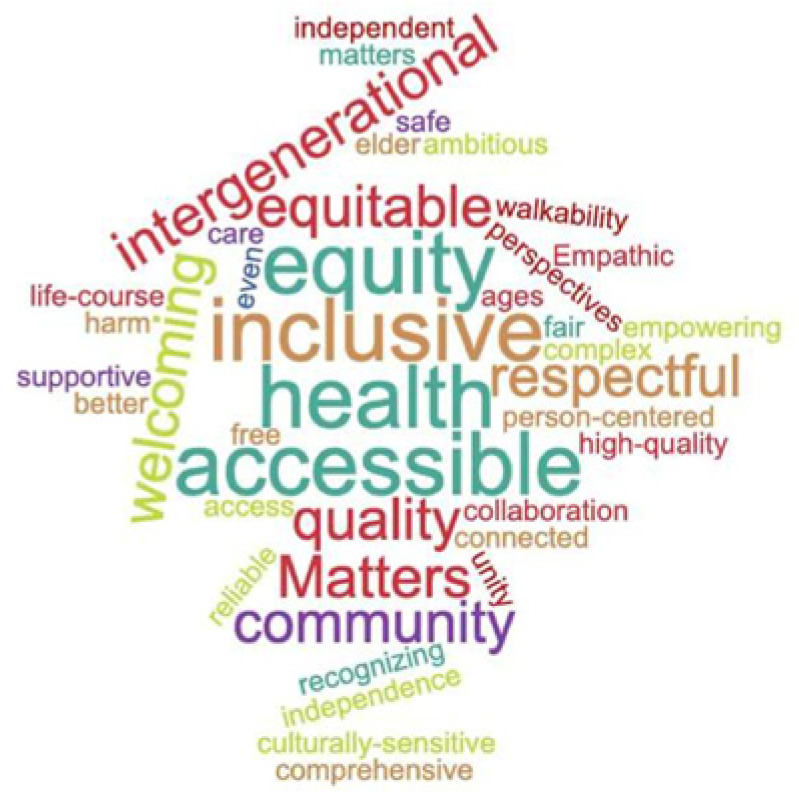
Expert Voices Were Asked to Describe Age-Friendly.

**Figure 3 geriatrics-08-00068-f003:**
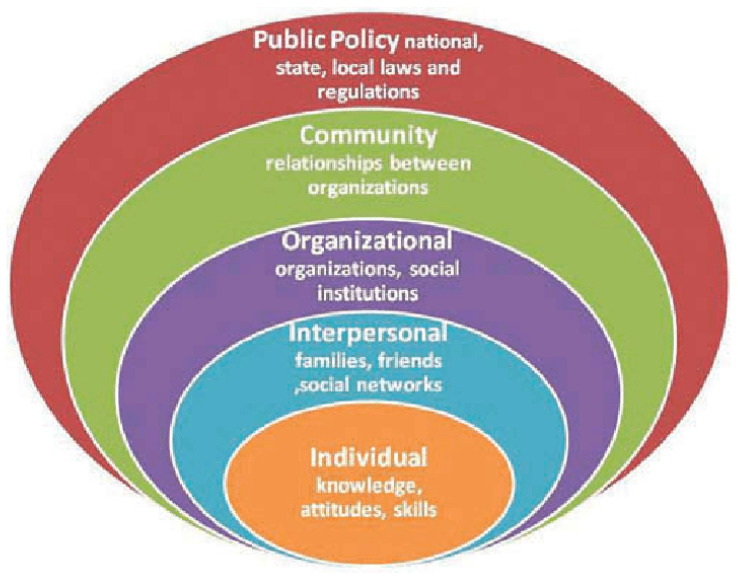
Social Ecological Model [9].

**Figure 4 geriatrics-08-00068-f004:**
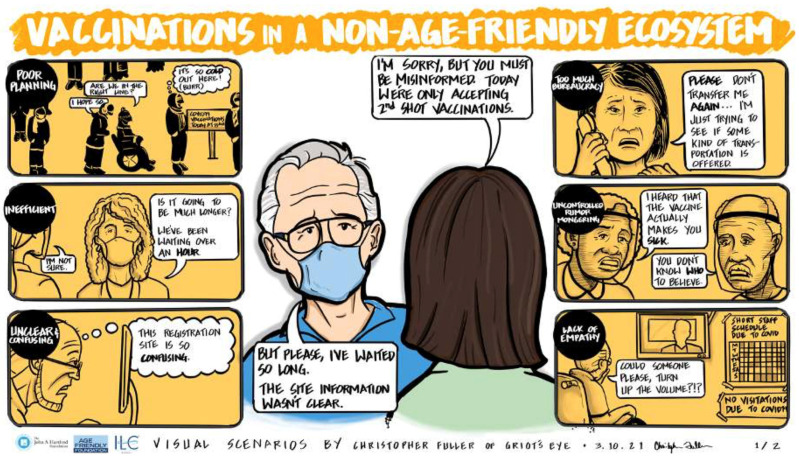
Visual Representation of Vaccine Journey Without a Functioning Age-Friendly Ecosystem.

**Figure 5 geriatrics-08-00068-f005:**
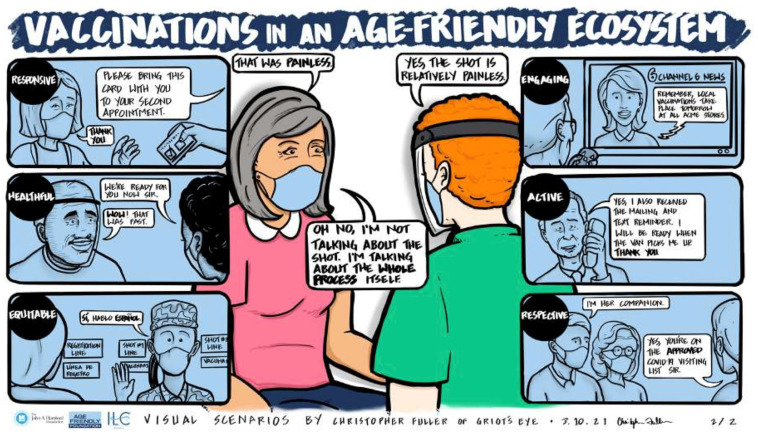
Visual Representation of Vaccine Journey With a Functioning Age-Friendly Ecosystem.

**Table 1 geriatrics-08-00068-t001:** Six Characteristics Comparing AFEs [2].

Responsive	The AFE is not a rigid framework. It should adapt and be responsive to the values and preferences of older adults identified through data collection and program assessments.
Equitable	AFEs should provide services that reach all older adults to mitigate inequity across all demographic factors.
Engaging	Engaging refers to including older adults in ways to bolster their quality of life and benefit society. This can be carried out in various ways, such as in an age-diversified workplace.
Healthful	AFEs should not solely focus on supporting the older adult but also empowering them to have agency in developing a healthful and high quality of life.
Active	Programs should focus on improving older adults’ mobility, which contributes to feelings of freedom and independence. Improvements to our built environments, such as walkability and reliable public transportation, are examples of ways to target this characteristic.
Respectful	Older adults should be respected and valued by society. Ageism led older adults to hide care needs due to fear of being seen as dependent and incapable. Redefining aging in a positive light can help improve older adults’ health outcomes.

**Table 2 geriatrics-08-00068-t002:** Various Expert Voices Highlights Number One Benefit that will be Achieved from AFEs.

Jody Shue, Executive Director of The Age Friendly Foundation asked attendees to answer the following question in the chat at the beginning of the meeting: What will be the number one benefit that will be achieved by organizations becoming part of an age-friendly ecosystem? Responses from attendees include:	
Erin Emery-Tiburcio Associate Professor Geriatric and Rehabilitation Psychology, Rush University Medical Center	Bridging traditional silos
Rani Snyder Vice President, Programs, The John A. Hartford Foundation	Greater understanding and connection
Judy Salerno, MD, MS President, NYAM	Improved quality of life for older persons
Nicole Brandt Professor, University of Maryland	Improved care delivery for older adults
Terry Fulmer President, The John A. Hartford Foundation	Better coordination and quality of life for older adults
Mark Kissinger President, K-Forward Consulting	Better care for families
Anne Doyle President, Lasell Village	Living a full, engaged, and purposeful life every day
Susan Reinhard Senior Vice President and Director, AARP Public Policy Institute & Chief Strategist, Center to Champion Nursing in America, AARP	Sharing Innovations
Lindsay Goldman Director, Healthy Aging, New York Academy of Medicine	More efficient use of resources and intellectual capital
Gretchen Alkema VP Policy and Communications, SCAN Foundation	Common Purpose
Anne Pohnert Director of Clinical Quality, CVS Health	Improved/enhanced human experience and equity
Christine O’Kelly Coordinator, Age Friendly University Global Network, Dublin City University	Broaden Participation
Kevin Little, PhD Improvement Advisor, Institute for Healthcare Improvement (IHI)	Greater impact, promote synergies
Melissa Batchelor, Ph.D., RN-BC, FNP-BC, FGSA, FAAN Associate Professor, George Washington University	Multi-sector connections to build the products, support and services need for healthy aging across the lifespan
Leslie Pelton Senior Director, Institute for Healthcare Improvement (IHI)	Older adults who are more engaged and empowered in their communities
Joan Weiss, PhD, RN, CRNP, FAAN Deputy Director, Division of Medicine and Dentistry, Health Resources and Services Administration	Improve healthcare and health outcomes for older adults
Megan Wolfe Senior Policy Development Manager, TFAH	Improved health and well-being for OAs!
Tim Driver President, The Age Friendly Foundation	Improved impact on the quality of experience for older adults
Rachel Roiland, PhD, RN Managing Associate, Duke-Margolis Center for Health Policy	Older adults feel more valued, respected and more connected to society
Terrie (Fox) Wetle Center for Gerontology and Healthcare Research, Brown University School of Public Health	Improved integration of older persons into society and better quality of life for us all
Randel Smith Patient Advocate	Better care for our aging population
Amy Berman Senior Program Officer, The John A. Hartford Foundation	The Age-Friendly Ecosystems initiatives promotes people and organizations working in different Age-Friendly domains to carry messages of the other domains and think how to integrate and accelerate efforts
Rebecca Stoeckle Vice President, Director, Private Sector Partnerships, Education Development Corporation	Systematizing care that is meaningful to older adults. These meetings are the embodiment of continuous communication, ensuring we are aligning goals and methods
Charles (Chuck) Pu Medical Director, Population Health, Mass General Brigham	Meaningful change starts with raising awareness and calling attention to a burning platform in a systematic organized framework

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
