# Peer review of "Age-Friendly Ecosystems: Expert Voices from the Field"

_geriatrics, 2023, doi:10.3390/geriatrics8040068_

Round 1

Reviewer 1 Report

This manuscript reports on a process to better define the concept of age-friendly ecosystems.  As a participant in the process, the description is accurate.  I offer a few specific comments to improve the manuscript.

In the Abstract, the first item "(1) Background" is not bolded as are the other numbered topics.  The caption for figure 2 reads "Expert Voices Use three words to Describe Age-Friendly", but there are four large words in the graphic.  This is confusing.

Table 2 has he first row in bold.  Why?

Lines 111-113 in Section 6 have quotations around "once we identify what we have in common, we will be able to overcome inertia, policy, and payment barriers, and fears of adding more layers to our work".  I do not see the need for these quotation marks, it is not clear just who is being quoted.

Header 7 "Expert Voices" should be edited.  The previous header (6) is "Expert Voices in Defining an Age-Friendly Ecosystem".   This is confusing.  

Author Response

Reviewer #1:

This manuscript reports on a process to better define the concept of age-friendly ecosystems.  As a participant in the process, the description is accurate.  I offer a few specific comments to improve the manuscript.

  1. In the Abstract, the first item "(1) Background" is not bolded as are the other numbered topics. 
  2. The caption for figure 2 reads "Expert Voices Use three words to Describe Age-Friendly", but there are four large words in the graphic.  This is confusing.
  3. Table 2 has the first row in bold.  Why?
  4. Lines 111-113 in Section 6 have quotations around "once we identify what we have in common, we will be able to overcome inertia, policy, and payment barriers, and fears of adding more layers to our work".  I do not see the need for these quotation marks, it is not clear just who is being quoted.
  5. Header 7 "Expert Voices" should be edited.  The previous header (6) is "Expert Voices in Defining an Age-Friendly Ecosystem".   This is confusing. 
  • Response #1: Thank you for your positive feedback and your helpful edits. In the abstract we have now corrected issues with the bolding, as well as the confusion related to the four large words in the graphic.
  • Response #2: Figure 2 title was updated to describe the word cloud more accurately.
  • Response #3: we have changed table 2 so that the first row is not in bold.
  • Response #4: In lines 111 through 113 we have adjusted the quotations and agree that they are not needed.
  • Response #5: Header 7 has now been edited to read “expert discussion”. We apologize for any confusion.

Reviewer 2 Report

The authors are to be commended for addressing age-friendly environments in this manuscript. It is a concept that is relevant to providers and care recipients alike. Even though the focus of the findings presented in this manuscript are well written and easy to read, the authors have simply omitted too much detail and rely on the readers to have already read or seek out their other published paper for more detailed information.

That being said, I strongly suggest the authors revisit the text and provide more detail. From the start, it is unclear what an AFE refers to, what "various sectors" are, what "silos" refer to, and who comprise the 40 international experts. It is all too vague. The names of the experts in the associated table are meaningless to me. What would be interesting is the roles they represent (e.g., healthcare coordinator, nursing academic, etc). The authors mention they are knowledgeable about all the sectors, but in what capacity? Without knowing more about the experts, why should the reader trust their input? 

Additional information should also be presented about the scientific method and analysis used to generate the expert voices. More quotes with a notation of the role of the expert who said them, would also be helpful.  

Lastly, an in-depth discussion of next steps seems to missing. How might the ecological model presented be interpreted differently for the sectors represented. For example, would a healthcare system envision or operationalize the AFE differently based on organizational type or structure? What role might culture play? How can leaders within the sectors move an AFE model forward in their community? If the authors addressed the possible ramifications of their work, the manuscript could become more thought-provoking and stimulate readers' thinking about AFEs. 

Author Response

Reviewer#2:

The authors are to be commended for addressing age-friendly environments in this manuscript. It is a concept that is relevant to providers and care recipients alike. Even though the focus of the findings presented in this manuscript are well written and easy to read, the authors have simply omitted too much detail and rely on the readers to have already read or seek out their other published paper for more detailed information.

That being said, I strongly suggest the authors revisit the text and provide more detail.

  1. From the start, it is unclear what an AFE refers to,
  2. what "various sectors" are, what "silos" refer to, and
  3. who comprise the 40 international experts. It is all too vague. The names of the experts in the associated table are meaningless to me. What would be interesting is the roles they represent (e.g., healthcare coordinator, nursing academic, etc.). The authors mention they are knowledgeable about all the sectors, but in what capacity? Without knowing more about the experts, why should the reader trust their input? 
  4. Additional information should also be presented about the scientific method and analysis used to generate the expert voices.
  5. More quotes with a notation of the role of the expert who said them, would also be helpful.  
  6. Lastly, an in-depth discussion of next steps seems to be missing. How might the ecological model presented be interpreted differently for the sectors represented. For example, would a healthcare system envision or operationalize the AFE differently based on organizational type or structure? What role might culture play? How can leaders within the sectors move an AFE model forward in their community? If the authors addressed the possible ramifications of their work, the manuscript could become more thought-provoking and stimulate readers' thinking about AFEs. 
  • Response #1: thank you for your commendation regarding this important topic. As you note in your review it is challenging and your good point about adding more content is very helpful. We have now provided more detail related to what an age friendly ecosystem.
  • Response #2: We have removed the term “silo” for clarification and identified the specific age-friendly sectors referenced in this paper.
  • Response #3: We had previously referenced a paper where we named the participating experts but have added an appendix to restate the composition of the group as well as their respective roles in who they represent.
  • Response #4: We note your recommendation that we add information about the scientific method and analysis on lines. Please see our additions and edits to the Methodology section.
  • Response #5: Thank you and we note your recommendation that we add more quotes with a notation of the role of the expert who said them.
  • Response #6: We have provided a more in-depth discussion of next steps and talked about how leaders can move the field forward. Thank you for this important idea.

Reviewer 3 Report

Dear authors, 

Interesting work.

My recommendations after reading it are some gaps:

Define what is an age-friendly ecosystem (AFE), make a further description.

And indicate a method followed for your article.

These are big absences and it would be interesting if you could add them.

Best regards,

Author Response

Reviewer #3:

Dear authors, 

Interesting work.

My recommendations after reading it are some gaps:

Comment #1:  Define what is an age-friendly ecosystem (AFE), make a further description.

Comment #2:  And indicate the method followed for your article.

These are big absences and it would be interesting if you could add them.

Best regards,

  • Response #1: Thank you. The definition is now on page #1.
  • Response #2: We note your recommendation that we add information about the scientific method and analysis and have added this content.

Round 2

Reviewer 2 Report

The authors made some positive changes to the manuscript that help make it suitable for publishing. The text could still withstand additional "tweaking" in terms of clarification, flow, and scientific rigor. These changes will help elevate the chances that this manuscript will be cited by future researchers. I strongly suggest the authors reread and edit for flow and remove repetitive text.  Other changes include the following

1. Please cite the reference to " An age-friendly ecosystem is defined as..."

2. Consider calling your "Experts" participants as they are participating in the study. They may have the common characteristic of being experts in their fields, but they are participants in the study. By constantly referring to them as experts, without defining their expertise or contributions to their field that gives them expert status, is not meaningful to the reader and overstates their value. Only a few references to participants as experts is needed. 

3. Change language in "(Expert participants are listed in Appendix A)" to (Participants and their professional roles are listed in Appendix A)

4. Wetle, 2020 is not listed in the reference list

5. Please cite "A Delphi Techniques... and Estimate-Talk-Estimate ..."

6. Please cite .."characterstics and supporting practices from the literature..."

7. Please elaborate further on the sentence that begins "Session 1, held in December 2020..." and references Figure 3. More needs to be said here. The reader is left hanging.

8. Eliminate Table 1. It is redundant as to what is stated in the text.

9. The paragraph beginning with "Following the first session..." is unclear. Take time to articulate the process used.

10. Add the date to the sentence beginning with "The goal of the second session.." You previously mentioned December 2020 for session one, what was the time frame for holding the second session?

11. I continue to dislike Figure 2 as it holds little meaning to the reader. Replace the person's name with their professional title (e.g., hospital administrator) so that the reader can get a sense of what the different professionals identified.

12. I would eliminate Table 3 and write the questions out in the text as done for the first session. Be consistent in presentation

13. A summary of all methodology should be provided at the beginning of the methods section. Move the sentence "All break out sessions were audio recorded..."

14. The sentence "The rationale for this cross-examination.." is important and lost in its current position. Consider moving it up in the text and revisiing paragraph.

15. Change the title of section 7 to Participant Viewpoints or Study Findings or ?

16. The Summary section remains weak. I would move the recent addition of text citing Kania and Kramer to the top after the term of collective impact is mentioned. There is currently  a lot of word fill in the Summary. Expand on the idea of a coordinating backbone agency and what that might look like. The community literature and the AFC literature identifes agencies such as the local AoAs as potential coordinators at the community level. What would a coordinating agency at the institutional level be? Give the reader something to think about.

Author Response

Enclosed you will find the authors' response.
